# The Impact of Time-Restricted Diet on Sleep and Metabolism in Obese Volunteers

**DOI:** 10.3390/medicina56100540

**Published:** 2020-10-14

**Authors:** Hyeyun Kim, Bong Jin Jang, A Ram Jung, Jayoung Kim, Hyo jin Ju, Yeong In Kim

**Affiliations:** 1The Convergence Institute of Healthcare and Medical Science, Department of Neurology, Catholic Kwandong University, International St. Mary’s Hospital, Incheon 22711, Korea; imkhy77@gmail.com; 2Department of Medical Business Administration, Daegu Hanny University, Gyeongsangbuk-Do 38610, Korea; jangbj@gmail.com; 3Department of Nutrition Management, Catholic Kwandong University, International St. Mary’s Hospital, Incheon 22711, Korea; greendevil23@ish.ac.kr; 4Department of Laboratory Medicine, Catholic Kwandong University, International St. Mary’s Hospital, Incheon 22711, Korea; lmkjy7@gmail.com; 5The Convergence Institute of Healthcare and Medical Science, College of Medicine, Catholic Kwandong University, Incheon 22711, Korea

**Keywords:** sleep, metabolism, time-restricted diet

## Abstract

*Background and objectives:* A time-restricted diet is one of the various ways to improve metabolic condition and weight control. However, until now, there have been few pieces of evidence and research to verify the methods and effectiveness of time-restricted diets on metabolic improvement and health promoting. We designed this study to make a healthy diet program and to verify the effectiveness of a time-restricted diet on general health, including sleep and metabolism, in healthy volunteers. *Materials and Methods*: This study was conducted in healthy adults who are obese but do not have related metabolic disease. Fifteen participants were recruited. Before and after this program, serologic tests including ketone level, questionnaires—daytime sleepiness evaluation such as the Epworth sleepiness scale and the Stanford sleepiness scale, the Korean version of the Pittsburgh sleep questionnaire index, STOP BANG to evaluate sleep apnea, the Hospital Anxiety and Depression Scale for emotion/sleep—and polysomnography (PSG) were conducted to evaluate the effects on sleep of the program. They were divided into two groups based on ketone levels that could reflect the constancy of participation in this study. We analyzed the before and after results of each group. *Results:* Fifteen participants (nine males and six females) completed this program without significant adverse events. Body weight after this program decreased to 78.2 ± 14.1 from 82.0 ± 15.6 kg (*p* = 0.539), and BMI decreased to 27.9 ± 3.8 from 29.3 ± 4.6 kg/m^2^ (*p* = 0.233). Weight loss was observed in 14 subjects except 1 participant. The results from questionnaires before and after this were not significant changes. They were classified into high/low-ketone groups according to the ketone level of the participants. In the results of the PSG, the apnea hypopnea index (25.27 ± 12.67→15.11 ± 11.50/hr, *p* = 0.25) and oxygen desaturation (18.43 ± 12.79→10.69 ± 10.0/hr, *p* = 0.004), which are indicators of sleep apnea, also improved in the high-ketone group, compared with the low-ketone group. Satisfaction interviews for this restricted diet program showed that 86% of the participants were willing to participate in the same program again. *Conclusion:* The time-restricted diet was successful in weight loss for a period of 4 weeks in obese participants, which did not affect the efficiency and architecture of sleep. In addition, successful weight loss and significant improvement of sleep apnea were showed in the high-ketone group. Further research is needed to demonstrate mechanisms for weight loss, sleep apnea, and time-restricted diets.

## 1. Introduction

The rate of obesity continues to increase in Korea, and analysis shows that the at-risk weight population has decreased, but the normal weight population did not increase; instead, it deteriorated into an obese population. Various methods, including the controlling of calories and protein, low carbohydrate diets, drinking sufficient water, more fiber consumption, and resistance exercises for weight loss, have been proposed as the interest in weight-control programs that do not harm health and improve the quality of life has increased. Intermittent fasting is one of the methods that has attracted attention in recent years. Several studies have reported that limiting the diet to intermittent fasting reduces cancer formation, slows the aging process, and improves stamina, body fat, and weight loss [1].

Forest products are any food materials derived from forestry. Most of the food ingredients of forest products are low-calorie, high in unsaturated fatty acids and fiber, richer than other food ingredients, and suitable as a food for weight control. In a recent study, forest products that lower insulin resistance have been reported, such as acorn and sago, making them highly useful for diets for patients with diabetes or metabolic diseases. In addition, mushrooms have been shown to have anti-obesity and antidiabetic properties [2,3,4].

Intermittent fasting is a dieting method that controls mealtimes. Low-calorie diets for weight control and time-restricted diets using them are commercially available. A time-restricted diet, in which meals were eaten within 10 h and fasting was done for the rest of the time, promoted weight loss in patients with metabolic syndrome [5]. Time-restricted diets are also known to reduce visceral fat, improve abdominal obesity, lower atherogenic lipids and glycated hemoglobin cholesterol, and control high blood pressure [6]. Modified fasting regimens or periodic very low-calorie diets have been shown to improve insulin resistance, reduce fasting blood sugar levels, and reduce weight [7]. However, information on their effects on sleep and metabolism is limited. A time-restricted feeding method using forest products, which focuses on healthy eating, may be an effective method for healthy weight control. We designed this study to develop a healthy low-calorie diet, using unsaturated fatty acids and fiber-rich forest product ingredients, and to test the effectiveness of time-restricted diets.

## 2. Methods

### 2.1. Participants

In order to perform research on weight/body fat loss and sleep and mental health through a time-restricted and calorie-limited diet, a study was conducted on obese adult volunteers without underlying diseases such as metabolic and mental disorders. The subjects were 15 adult males and females with a body mass index over 25 and between 20 and 50 years old. Based on the study participation date, those who had major surgery within 3 months or were allergic to certain foods were excluded. Through advertisements in and out of the hospital, subjects who participated in this clinical research were recruited on a first-come, first-serve sequential basis. Before and after the program, physical examinations were conducted to measure parameters such as height, weight, body mass index, and blood pressure, and questionnaires were completed related to emotion and sleep.

Each participant’s sleep and emotional status was assessed using the Epworth sleepiness scale (ESS), the Stanford sleepiness scale (SSS) to evaluate daytime somnolence, the Korean version of the Pittsburgh Sleep Questionnaire Index (PSQI) for sleep quality investigation, insomnia severity index for measuring the severity of insomnia symptoms, the Snoring, Tiredness, Observed apnea, high blood Pressure, BMI, Age, Neck circumference, and male Gender (STOP BANG ) to evaluate sleep apnea, and the Hospital Anxiety and Depression Scale (HADS) before and after this program [8,9,10,11,12]. Polysomnography (PSG) was conducted before and after forest therapy, but baseline PSG was conducted one week before participation in the sleep lab of the research institute. PSG was performed using a digital PSG machine (Nox A1, Nox Medical Inc., Reykjavik, Iceland). The following variables were monitored: electroencephalogram (EEG; C3-A2, C4-A1, O2-A1, O1-A2), right and left electro-oculogram, submental, both anterior tibialis electromyograms, electrocardiogram, airflow (pressure cannula and thermistor), respiratory effort (piezoelectric bands), oxyhemoglobin saturation (SaO_2_), and snoring. This was performed by an experienced sleep technician, and the scoring and staging were performed by a sleep physician. The participants had blood sampling before and after the program for a general evaluation that included common blood cells, glucose, the enzymes aspartate transaminase and alanine transaminase (alanine aminotransferase), a lipid profile including total cholesterol, LDL-cholesterol, HDL-cholesterol and triglyceride, and fasting glucose/insulin. Ketone values were measured four times at weekly intervals.

The appropriate ethics review board approved this study design (# IS19OISI0046, 2019.08.12). All patients gave their consent to participate in the study.

### 2.2. Statistical Analysis

All values from this study are presented as the mean ± standard deviation. The difference between the mean values of the variables including the results of the serology, questionnaire, and PSG before and after the time-restricted diet was analyzed using the paired *t*-test. A *p* value of less than 0.05 was considered statistically significant. All analyses were conducted using IBM Statistics SPSS software (version 26.0 SPSS, Inc., Chicago, IL, USA).

### 2.3. Diet

The diet for this program was formulated in consultation with a dietitian. Two meals per day were provided for a total of four weeks: the same for the first two weeks, and a different composition for the 3rd and 4th weeks (Table 1). Time-restricted feeding was conducted over 4 weeks. In the 1st and 3rd week, the diet was prepared according to a low carbohydrate diet; the 4th week was a stage of transition to a regular diet, and the carbohydrate ratio was increased to form the transitional diet. The total calories per day were 1350 kcal. The two meals were made into lunch boxes and provided to the participants. Lunch boxes included menus using forest products such as mushroom salad, bellflower salad, acorn cake salad, fried fern, and chestnut salad. Figure 1 shows sample lunch boxes provided to participants. There were 23 kinds of forest products used: ginkgo, chestnut, acorn, hemp, mushrooms, burdock, bellflower, fern, forest herbs, bamboo shoot, walnut, pine nut, peanut, omija, and mulberry.

Participants were asked to eat within 8 h from 12 p.m. to 8 p.m. For the rest of the time, they kept fasting; for hunger and thirst, they were allowed to drink water or *deodeok* tea as much as possible. Intermittent coffee was allowed. We called the participants every morning to make sure they had eaten only the supplied meal box and kept to the designed time according to the program, and encouraged them to faithfully carry out the program on that day. It was recommended that the time-restricted diet should be observed on weekends as well.

## 3. Results

Fifteen obese volunteers (nine males and six females) participated in this study and completed it without failing. The average participant age was 36.8 ± 8.44 (mean ± standard deviation) years old, and the baseline BMI was 29.3 ± 4.62. Table 2 shows the changes in body weight, BMI, body fat, and muscle mass before and after this study. Body weight after this program changed from 82.0 ± 15.6 to 78.2 ± 14.1 kg (*p* = 0.539), and BMI decreased from 29.3 ± 4.6 to 27.9 ± 3.8 without statistical significance (*p* = 0.233). Weight loss was observed in 14 of the 15 subjects.

During the study period, if the participants consumed more than two meals per week that where not provided in the lunchbox, or if the participant ate food during the fasting time (between 12 p.m. and 8 p.m.) more than twice per week, this was defined as inadequate participation. Six of the participants did not adhere to our instructions, which was also represented with their low ketone levels. They measured ketone values at weekly intervals. Based on the highest value of ketone levels, participants were divided into a high-ketone group that exceeded 1 mM and a low-ketone group that did not exceed 1 mM. There were six subjects in the low-ketone group and nine subjects in the high-ketone group. Figure 2 shows the changes in ketone levels performed at 1-week intervals during the study period in the high- and low-ketone groups. The high-ketone group generally showed a pattern of rising at week 2 and then falling, whereas the low-ketone group showed an irregular pattern with no pattern of change (Figure 2). After the time-restricted diet, body weight, body fat mass, and BMI showed significant decrease in the high-ketone group that participated sincerely (Table 3). Table 4 shows the results of blood tests in each group. In particular, during the blood test of the high-ketone group, significant changes in insulin level (*p* = 0.006) and an improved trend of insulin resistance as a measured homeostatic model assessment for insulin resistance (HOMA-IR) (*p* = 0.052) were confirmed.

The results from the questionnaires before and after this program, including daytime sleepiness evaluation with the Epworth sleepiness scale, the Stanford sleepiness scale, the Korean version of the Pittsburgh Sleep Questionnaire Index, STOP BANG to evaluate sleep apnea, and the Hospital Anxiety and Depression Scale did not show significant changes. The results of the analyses of the sleep study conducted before and after participation in the program are shown in Table 5. There were no changes in sleep structures before or after the program in Table 6. However, in the high-ketone group, two indicators of sleep apnea improved: the apnea hypopnea index decreased (from 25.27 ± 12.67 to 15.11 ± 11.50, *p* = 0.25) and oxygen desaturation decreased (from 18.43 ± 12.79 to 10.69 ± 10.69, *p* = 0.004) (Figure 3).

The satisfaction score when the meal was very unsatisfying was rated at 0 points; very satisfying was 5 points. The average score for the overall diet was 4.1 ± 0.7 points, and the score for the forest products dishes was 3.9 ± 0.8 points. Satisfaction interviews for this restricted feeding program showed that 86% of the participants were willing to participate in the same program again.

## 4. Discussion

This study investigated a universally applicable weight-control diet that is effective and does not cause unexpected medical problems. In many cases, weight-control failure is a diet-control failure. In particular, it is not easy to adjust time limits, calorie restrictions, and dietary carbohydrate regulations for ordinary people familiar with only a general diet. We developed meals by focusing on the following three things in this study: (1) developing a variety of diets using healthy forest products, (2) devising an effective time-restricted diet method, and (3) providing adequate calories while reducing the ratio of carbohydrates. Although the number of participants in the study was not large, it was concluded that this program was completed successfully: the initial goal of this study was to complete the four-week program without any participant fallout.

In this study, forest products were used as ingredients for lunch boxes. *Codonopsis lanceolata*, also called *deodeok* root tea, was provided to prevent hunger and dehydration during the fasting time of the participants. Salads and chicken breast are the main ingredients in commercial diet lunch boxes common in Korea. The prices of forest products are higher than that of other food ingredients, and little is known about their utilization. In a study using mushrooms, one of the representative clinical products, mushrooms had anti-obesity and antidiabetic properties [2,3,4]. There are not many dishes that use mushrooms as the main ingredient, but we obtained high satisfaction ratings from the participants for the mushroom-containing dishes. The effects of various forest products cannot be generalized, but this study shows that they qualified as low-calorie dietary material rich in unsaturated fatty acids and fiber. Using forest products in a weight-control diet might make it possible to introduce new healthy food to people who are tired of existing weight-control diets.

Meals are the basis of the daily cycle, which can affect sleep cycles. Recent studies indicate that living a healthy lifestyle, with sleeping and eating consistent with circadian rhythms, can be maintained [6]. Intermittent and regular fasting offer a wide range of benefits, from disease prevention to enhanced treatment. Similarly, a time-restricted diet in which food consumption is limited to a specific time of day can strengthen all existing benefits by allowing the daily fasting period to last 12 h or more. Several studies have suggested fasting-related interventions are a feasible, effective, and inexpensive treatment with the potential to promote health [9,10,11]. In addition, it has been reported through animal experiments that limiting the nighttime diet normalizes clock genes [12]. It is said that the daily sleep cycle and a time-restricted diet work in synergy [13]. In addition, it has been reported that a time-restricted diet has a positive preventive effect on cardiovascular and metabolic diseases [14]. In this study, it is hard to see if this is a short-term weight loss effect or whether this is the effect of a time-restricted diet. However, there are a few things that are clear: the first is that the effects of short-term weight loss have a positive effect on sleep apnea, and the second is that changes in chronobiology caused by a time-restricted diet do not affect sleep structure and sleep duration. However, it is necessary to confirm how each factor affects through future research.

In this study, diet schedules were allowed for 8 h a day, lunch was served at noon, and dinner was from 6–7 pm. Participants took appropriate amounts of alternating pure water or tea during fasting time. After waking up in the morning, no meals were provided until lunchtime, and water or tea was served. Participants said that it was not easy to endure fasting during the first week, but from the second week onwards, they became accustomed to fasting and had no significant problems associated with fasting. In telephone interviews six months after the program ended, five people, one-third of all participants, said they were still skipping breakfast and staying on the same mealtime schedule. The reason they keep the same schedule as the program is that this schedule might stop them from gaining weight and helps them to maintain a healthier daily life. The time-restricted diet schedule proposed in this study proved effective for weight control and had no side effects of special medical problems for healthy adults.

Despite the new meal schedule, the structure and efficiency of sleep before and after the program did not show significant changes. Some patients subjectively seemed to be going to sleep more deeply and said that the time to sleep was faster. However, the results of the polysomnographic data did not differ significantly. According to sleep studies conducted during the Ramadan period, which has more thorough fasting, an increase in sleep latency and a decrease in deep sleep and REM sleep were reported [15]. During Ramadan, eating occurs after sunset, which is a more time-restricted feeding than in our program. It is thought that the schedule of 16 h fasting and 8 h meals does not significantly affect sleep health. In this study, participants lost weight while controlling calories and carbohydrates, and, as a result of the weight loss, the degree of sleep apnea showed significant improvement. This was an unexpected positive result.

Ketones are one of the indicators of sleep regulation, and during sleep, ketone bodies in the cerebral fluid increase [16]. Sleep deprivation makes increased serum concentration of ketone bodies and increased expression of ketogenesis-related genes in the brain. The level of ketone bodies and their metabolism in the brain could affect sleep homeostasis [17]. It is not clear whether increased ketones play a protective role in sleep or a promotional role through several studies, but it could be seen that it has a positive effect on sleep. In this study, the rate of increase in slow wave sleep (stage N3) in the high-ketone group was greater than that of the low-ketone group. Although statistical significance was not shown, further study is needed to determine whether the greater increase in slow wave sleep, that is, deep sleep in the high-ketone group, was the result of the sleep-enhancing effect of ketones shown in previous studies on the relationship between ketones and sleep.

The research revealed that developing lunch boxes using forest products and time-restricted feeding schedules does not change sleep patterns, but improves insulin resistance. However, there are some limitations. First, all participants in the 4-week program were evaluated by their ketone level, but the basis for this is weak. While maintaining low calories and low carbohydrates, ketone levels may increase, but this is due to intraday changes, exercise-intensity effects, and individual differences. In order to clarify this, measuring changes in ketone levels at shorter intervals might be helpful. Second, it is difficult to judge the additional effects obtained by using forest products as food ingredients. It was hypothesized that the use of forest products might have an additional effect, as it was completed without any participants dropping out for 4 weeks, and there were high satisfaction ratings for food made from forest products. However, in order to prove this, a comparative study should be conducted by comparison with an ordinary lunch box that does not use forest products. Third, the chronotypes of the subjects were not reflected in this study. Chronotypes during a time-restricted diet could be divided into morning and evening types. This program would have been more adaptable in the evening types because of the restricted mealtime. Chronotype could influence time-restricted diets because it is an important factor to determine the beginning and end of the diet. Depending on the chronotype, the success or failure of these dietary restriction studies can vary. Follow-up studies addressing these limitations might be able to resolve any new questions we have gained through this study.

## 5. Conclusions

The time-restricted diet using forest products was successful in weight control for a period of 4 weeks, but did not change the structure of sleep. The result of weight loss showed a positive effect on improving sleep apnea for 4 weeks. Through this study, we clinically investigated the positive effects of a time-restricted diet, and further studies are needed to establish a basis for its mechanism.

## Figures and Tables

**Figure 1 medicina-56-00540-f001:**
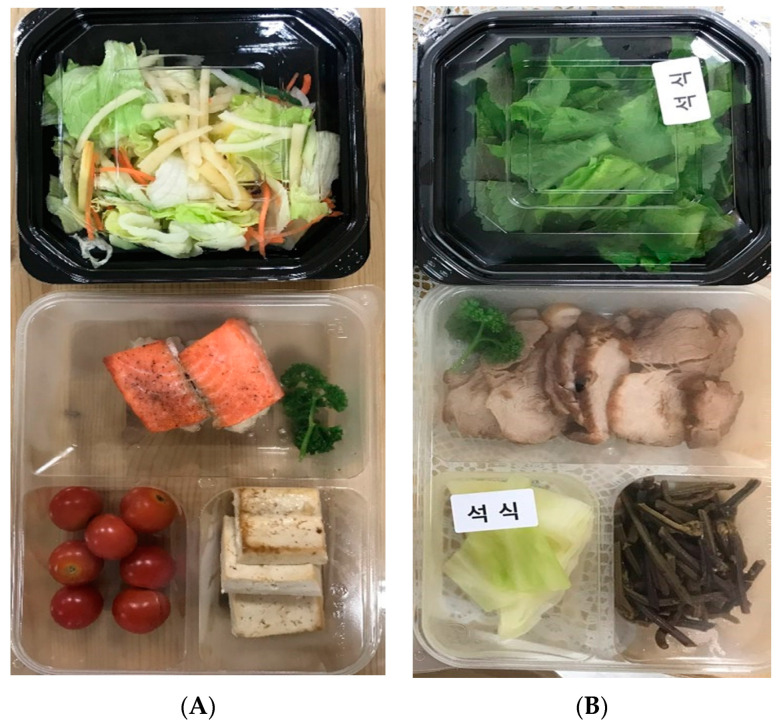
Example of lunch boxes. (**A**) Smoked salmon steak, cherry tomato, grilled tofu, black sesame dressing and bellflower salad. (**B**) Pork meat, stir-fried bracken, lettuce sesame leaf salad, and steamed cabbage.

**Figure 2 medicina-56-00540-f002:**
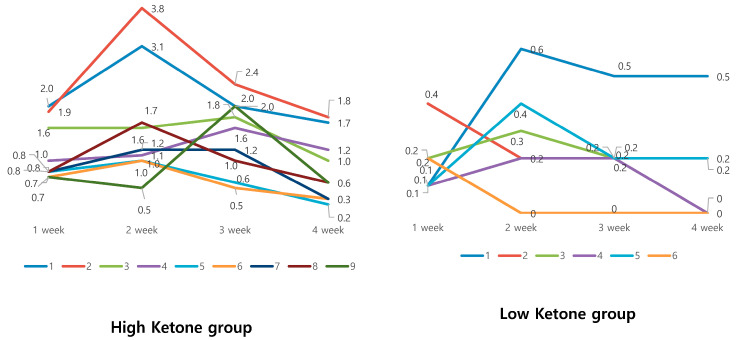
Ketone changes between high- and low-ketone group. Participants were divided into high-ketone groups that exceeded 1 mmol/L and low-ketone group that did not exceeded 1 mmol/L.

**Figure 3 medicina-56-00540-f003:**
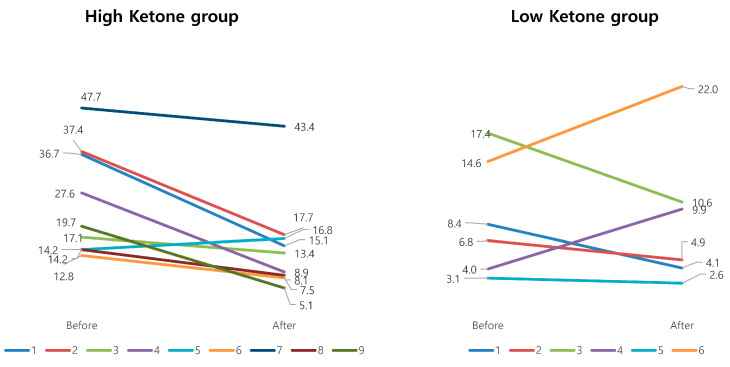
Apnea hypopnea index changes between high- and low-ketone groups. Apnea hypopnea indexes in the high-ketone group showed the tendency to improved changes, compared with those in the low-ketone group.

**Table 1 medicina-56-00540-t001:** Dietary compositions.

	Nutrition Composition (g)			% kcal		
	Carbohydrate	Protein	Fat	Carbohydrate	Protein	Fat
1, 2 weeks	26.8	88.5	90.1	8	28	64
3 weeks	32.5	90.6	97.6	9	26	65
4 weeks	65.4	85.2	90.0	18	24	58

**Table 2 medicina-56-00540-t002:** Basic characteristics of participants (*n* = 15).

	Before Program	After Program	*p*
Body weight (kg)	82.0 ± 15.6	78.2 ± 14.1	0.539
Body Mass Index (kg/m^2^)	29.3 ± 4.6	27.9 ± 3.8	0.233
Body fat mass (kg)	29.5 ± 8.7	26.1 ± 7.6	0.285
Body muscle mass (kg)	29.4 ± 7.1	29.0 ± 6.8	0.870

**Table 3 medicina-56-00540-t003:** Metabolic changes between high/low-ketone groups (*n* = 15).

	Low Ketone Group (*n* = 6)		High Ketone Group (*n* = 9)		*p*
	Before	After	Before	After	
Body weight (kg)	75.23 ± 75.14	73.90 ± 14.66	86.43 ± 15.03	81.13 ± 13.71	0.006
Body Mass Index (kg/m^2^)	27.33 ± 1.64	26.93 ± 1.90	30.61 ± 5.56	28.62 ± 4.71	0.008
Body fat mass (kg)	28.40 ± 2.42	27.22 ± 2.71	30.20 ± 11.23	25.42 ± 9.69	0.007
Body muscle mass (kg)	25.83 ± 8.70	25.65 ± 8.06	31.76 ± 4.98	31.29 ± 5.03	0.522

**Table 4 medicina-56-00540-t004:** Changes in serologic variables between high/low-ketone groups (*n* = 15).

	Low Ketone Group (*n* = 6)		High Ketone Group (*n* = 9)		*p*
	Before	After	Before	After	
AST	20.50 ± 3.45	23.00 ± 6.23	31.11 ± 16.10	31.89 ± 19.19	0.600
ALT	19.83 ± 9.41	20.17 ± 7.22	62.33 ± 75.22	45.33 ± 47.42	0.192
BUN	11.55 ± 3.16	12.50 ± 4.61	14.74 ± 2.88	14.27 ± 2.79	0.521
Creatine	0.56 ± 0.10	0.58 ± 0.11	0.76 ± 0.15	0.83 ± 0.16	0.223
γ-GTP	35.00 ± 13.60	24.00 ± 10.77	42.67 ± 34.44	19.89 ± 13.60	0.257
ALP	69.17 ± 20.17	70.17 ± 15.68	63.78 ± 15.68	66.22 ± 18.55	0.803
Total Cholesterol	173.33 ± 31.94	169.17 ± 21.10	186.22 ± 33.46	180.56 ± 45.36	0.910
LDL Cholesterol	105.00 ± 24.23	99.67 ± 16.08	116.89 ± 25.30	110.44 ± 32.01	0.909
TG	119.50 ± 55.39	129.33 ± 73.55	153.67 ± 67.54	172.89 ± 231.20	0.934
Glucose	95.67 ± 29.04	104.33 ± 5.99	115.56 ± 45.12	96.33 ± 8.03	0.175
Insulin	8.70 ± 1.72	8.08 ± 1.31	15.37 ± 5.53	8.14 ± 5.53	0.006
HOMA IR	2.27 ± 0.58	2.09 ± 0.42	4.79 ± 3.98	2.00 ± 1.49	0.052

AST—Aspartate transaminase; ALT—Alanine transaminase; BUN—Blood urine nitrogen; r-GTP—Gamma Glutamyl Transpeptidase; ALP—Alkaline phosphatase; TG—Triglyceride; HOMA-IR—Homeostatic Model Assessment for Insulin Resistance.

**Table 5 medicina-56-00540-t005:** Questionnaire results before and after high/low-ketone group (*n* = 15).

	Low Ketone Group (*n* = 6)		High Ketone Group (*n* = 9)		*p*
Variables	Before	After	Before	After	
PSQI-K ^1^	8.00 ± 3.46	6.50 ± 1.22	7.11 ± 2.37	5.11 ± 0.78	0.759
STOP BANG	2.50 ± 1.64	2.50 ± 1.64	3.00 ± 1.00	3.00 ± 1.00	-
SSS ^2^	2.17 ± 0.98	2.83 ± 1.17	2.56 ± 0.23	2.67 ± 0.50	0.256
ESS ^3^	9.00 ± 3.46	10.00 ± 2.19	9.11 ± 4.59	8.89 ± 4.91	0.465
HADS ^4^ Anxiety	4.67 ± 2.34	6.33 ± 3.93	6.78 ± 2.54	6.44 ± 2.55	0.225
HADS Depression	3.33 ± 1.86	4.50 ± 1.38	4.67 ± 3.32	4.11 ± 2.37	0.082
ISI ^5^	6.17 ± 3.87	5.00 ± 1.41	7.75 ± 1.83	5.38 ± 2.97	0.529

^1^ Pittsburg Sleep Quality Scale-K, ^2^ Stanford Sleepiness Scale, ^3^ Epworth Sleepiness Scale, ^4^ Hospital Anxiety and Depression Scale, ^5^ Insomnia Severity Index.

**Table 6 medicina-56-00540-t006:** Polysomnographic findings between the high/low-ketone groups (*n* = 15).

	Low Ketone Group (*n* = 6)		High Ketone Group (*n* = 9)		*p*
	Before	After	Before	After	
Total Sleep Time (min)	425.53 ± 37.96	415.32 ± 97.12	428.39 ± 51.76	442.87 ± 42.44	0.547
Latency to sleep onset	11.30 ± 3.91	27.02 ± 21.57	28.23 ± 34.01	27.61 ± 31.93	0.295
Latency to REM onset	90.03 ± 30.02	81.82 ± 52.77	148.90 ± 129.27	69.80 ± 61.56	0.220
Sleep Efficiency (%)	90.40 ± 7.57	89.17 ± 6.19	89.09 ± 8.99	86.50 ± 6.07	0.776
Sleep stage N1 (%)	1.80 ± 0.49	1.48 ± 0.33	3.49 ± 4.49	2.63 ± 1.26	0.775
Sleep stage N2 (%)	54.23 ± 8.97	52.83 ± 10.90	59.92 ± 13.38	55.50 ± 6.49	0.683
Sleep stage N3 (%)	19.73 ± 8.35	21.42 ± 5.80	15.51 ± 3.91	19.09 ± 8.67	0.699
Sleep stage REM (%)	24.25 ± 3.69	24.27 ± 5.65	21.10 ± 10.00	22.57 ± 5.41	0.745
Wake After Sleep Onset	45.40 ± 41.60	44.92 ± 35.97	51.56 ± 42.63	69.54 ± 41.63	0.472
AHI	9.05 ± 5.78	9.02 ± 7.13	25.27 ± 12.67	15.11 ± 11.50	0.025
PLMSi	4.78 ± 7.86	2.85 ± 5.28	30.90 ± 49.32	19.57 ± 44.70	0.752
Oxygen desaturation index	4.07 ± 3.02	5.87 ± 6.59	18.43 ± 12.79	10.69 ± 10.69	0.004

REM—Rapid eye movement; AHI—Apnea Hypopnea Index; PLMSi—Periodic limbs movement during sleep index.

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
