# Peer review of "The Impact of Time-Restricted Diet on Sleep and Metabolism in Obese Volunteers"

_medicina, 2020, doi:10.3390/medicina56100540_

Round 1

Reviewer 1 Report

This paper evaluated the methods and effectiveness of the time-restricted diet which is considered a useful strategy to ameliorate metabolic condition, weight control and health promotion. Interestingly, authors proposed time- restricted diet as been successful in weight loss in obese participants, without affecting the efficiency and architecture of sleep.

Introduction

  • The authors declared that various methods of weight loss have been proposed for weight control programs. I think that the authors could give some examples.
  • The Authors discuss the low-calorie properties of the forest products.  Could the authors define the meaning of “forest products”? What kind of foods does this group includes?
  • I would suggest to discuss more on the effect of time-restricted diet and sleep on cardiometabolic parameters and obesity. Please, consider the following papers: DOI: 1012997/jla.2020.9.1.140; 10.3389/fgene.2018.00514; 10.1016/j.numecd.2018.04.002

Methods

  • The authors reported that several scales have been used to evaluate sleep and emotional status. For instance, they cited the Epworth sleepiness scale,Stanford sleepiness scale, the Korean version of the Pittsburgh Sleep Questionnaire Index, theSTOP BANG and the Hospital Anxiety and Depression Scale. Should the authors provide some details about these scales?
  • I think that the authors should specify what statistical software has been used to perform statistical analyses.

Diet

  • Table 2 reported the characteristics of study participants. I think that would be really helpful to make the table more effective for the reader and easier to interpret, maybe using a figure. Moreover, this table was not mentioned in any paragraph.

Results

  • I suggest to verify the tables’ numbering. Why the authors began to number starting by second table?
  • Why Figure 1 was not mentioned in any paragraph?

Conclusions

  • I suggest that the Conclusions should be more incisive and effective.

Author Response

Dear sir

Thank you for your kind, great analytical comment.

#1. Introduction

The authors declared that various methods of weight loss have been proposed for weight control programs. I think that the authors could give some examples.

à Thanks for your comment. I added the content as below.

“Various methods including controlling of calories and protein, low carbohydrate diet, drink sufficient water, more fiber eats and resistance exercises of weight loss have been proposed as the interest in weight control programs that do not harm health and improve the quality of life has increased.”

#2. The Authors discuss the low-calorie properties of the forest products.  Could the authors define the meaning of “forest products”? What kind of foods does this group includes?

à Thank you for your comments. I added more documents like this. We have once again mentioned the forest products used in the "method" section. We used all 23 types of forest products to make this study meal box.

“Forest product is any food material derived from forestry. Most of the food ingredients of forest products are low-calorie, high in unsaturated fatty acids and fiber, richer than other food ingredients, and suitable as a food for weight control. In a recent study, forest products that lower insulin resistance have been reported, such as acorn and sago, making them highly useful for diets for patients with diabetes or metabolic diseases. In addition, mushrooms have been shown to have anti-obesity and anti-diabetic properties.2-4

“There were 23 kinds of forest products used: ginkgo, chestnut, acorn, hemp, mushrooms, burdock, bellflower, fern, forest herbs, bamboo shoot, walnut, pine nut, peanut, omija, and mulberry.”

I would suggest to discuss more on the effect of time-restricted diet and sleep on cardiometabolic parameters and obesity. Please, consider the following papers: DOI: 1012997/jla.2020.9.1.140; 10.3389/fgene.2018.00514; 10.1016/j.numecd.2018.04.002

  • We have described additional content based on your comments. We have also added references that you recommend.
  •  

“Intermittent fasting is a dieting method that controls mealtimes. Low-calorie diets for weight control and time-restricted diets using them are commercially available. Modified fasting regimens or periodic very low calorie diets have been shown to improve insulin resistance, reduce fasting blood sugar levels, and reduce weight.5

Methods

The authors reported that several scales have been used to evaluate sleep and emotional status. For instance, they cited the Epworth sleepiness scale,Stanford sleepiness scale, the Korean version of the Pittsburgh Sleep Questionnaire Index, theSTOP BANG and the Hospital Anxiety and Depression Scale. Should the authors provide some details about these scales?

  • According to your recommendation, we have added a description of the questionnaire used in the'Method' section as follows.

“Each participant’s sleep and emotional status was assessed using the Epworth sleepiness scale (ESS), the Stanford sleepiness scale (SSS) to evaluate the daytime somnolence, the Korean version of the Pittsburgh Sleep Questionnaire Index (PSQI) for sleep quality investigation, insomnia severity index for measure the severity of insomnia symptoms, the STOP-Bang (to evaluate sleep apnea), the Hospital Anxiety and Depression Scale (HADS), and PSG before and after the this program.6-10

I think that the authors should specify what statistical software has been used to perform statistical analyses.

Diet

Table 2 reported the characteristics of study participants. I think that would be really helpful to make the table more effective for the reader and easier to interpret, maybe using a figure. Moreover, this table was not mentioned in any paragraph.

  • Thanks for your point. I was not aware that Table 2 was not mentioned in the text. I mentioned further in the text.

“Table 2 shows the changes in body weight, BMI, body fat mass mass, and muscle mass before and after this study.”

Results

I suggest to verify the tables’ numbering. Why the authors began to number starting by second table?

Why Figure 1 was not mentioned in any paragraph?

à In the result section, more descriptions were added for the contents presented only as pictures.

“Figure 1 shows the changes in ketone levels performed at 1-week intervals during the study period in the high and low ketone groups. The high ketone group generally showed a pattern of rising at week 2 and then falling, whereas the low ketone group showed an irregular pattern with no pattern of change (Figure 2).”

Conclusions

I suggest that the Conclusions should be more incisive and effective.

  • I agree with your comment. The sentence that presents a more clear conclusion has been modified as follows.

The time-restricted diet using forest products was successful in weight control for a period of 4 weeks and did not change the structure of sleep. The result of weight loss showed a positive effect on improving sleep apnea for 4 weeks. Through this study, we clinically investigated the positive effects of the time restricted diet, and further studies are needed to establish a basis for its mechanism.

Reviewer 2 Report

This interesting article by Kim et al investigates the effect of a time-restricted and low-calorie diet with forest products on weight loss and sleep outcomes. They used many different methods to assess blood values and sleep. Although interesting, there are some doubts on the analysis structure and reporting of the findings. It is a complicated intervention, changing both diet timing, diet content, and calorie amounts at the same time. In general, I would highly recommend not to split the data on ketone levels, as the scientific basis for this is weak, as you also mention in your discussion.

Introduction

The introduction should include more information regarding the effects of diets on sleep. Especially as you include several different facets (diet quality, timing, and calorie restrictions), which in my opinion are not sufficiently introduced, which makes the choice for this study and setup unclear. Regarding f.e. eat timing, several important papers should be discussed in light of your study: 10.1016/j.cmet.2015.09.005 , 10.1016/j.cmet.2019.11.004

Methods

“Under 20-50 years old” should be changed to “between 20-50 years old”.

What p-value did you consider to be statistically significant?

In the diet section, please describe what rules the participants needed to follow with regards to the eating time window. Right now, the only focus lays on the low calorie-diet. Also, how did you confirm that the participants adhered to your instructions regarding food amount and food timing?

Regarding the polysomnography, where did the participants sleep? And who scored the polysomnography?

Results

You mention multiple times that the participants´ body weight and BMI decrease. However, you show no significant differences in these variables when analyzing these measures before and after the intervention. This is certainly a stretch of interpretation, as your results and p-values show  no differences (p=0.539 and p=0.233) Thus, there is no significant weight loss in these participants after the intervention. Please adjust.

You say you base the cut off for the ketone values on the highest value of ketone levels. Could you elaborate a bit more on this? Did you take half of the highest value as a cut off? Was this the highest mean value of one participant or of all participants? This is unclear.

In table 3 and 4, is it correct that you only show the statistical outcomes for the high ketone group? Why did you do this?

Although there are indeed significant changes in insulin in the high ketone group following the intervention (p=0.006), for the insulin resistance (as measured with HOMA), there is only a trend visible (p=0.052), which should be clarified.

For all tables, please provide an abbreviation list in the description under each table. Also, please specify if these values are mean ± SD, median ± SEM, or ..?

Why did you choose to show each participant´s ketone changes in figure 1 and 2? Given the mean values are already available in the tables, the added value of these figures is unclear to me.

Please report the outcomes of the Epworth sleepiness scale, Stanford sleepiness scale, and the Korean version of the Pittsburgh Sleep Questionnaire Index, even if they are not significantly different. Also, why did you choose here not to show the overall group findings? As you mention later in your discussion, splitting participants on their ketone body levels is maybe not the right measure to evaluate differences of such a multi-layer intervention.

Although you didn´t evaluate participant´s chronotype, what were their bed and wake times? This could be highly affected by the eating onset and offset, and vice versa. You could use this f.e. from the PSQI. This should also be discussed in the discussion.

Discussion

You are testing three different facets of diet (eating time restriction, dietary content (forest products), and calorie restriction) at the same time. How do you know where the (lack of) effects come from? This should be thoroughly discussed. This will also make the comparison to other studies difficult, but this should be at least attempted.

The conclusions written here are too bold, especially given non-significance.

If you choose to keep the ketone-split in your paper, the discussion really needs to include a section where the differences found in high-ketone group but not low-ketone group are discussed.

Every person has a different calorie need, also based on f.e. exercise. How could this have impacted the results, as all participants ate 1350 kcals? Or where the participants matched on energy expenditure levels?

Author Response

Dear sir.

I appreciate your detailed and kind reviews on this manuscript.

I totally agree with your comments. However, there is some limitation due to human study and ethical issues. It was not easy to control food other than the provided lunch box, and participants sometimes ate other foods, especially on weekends. Although it was described as a subgroup based on ketone values, it was actually divided into two groups based on the royalty of participation in this study. There is some misunderstanding as a description issue, so the description for this part has been corrected and the subgroup name has been corrected.

Introduction

The introduction should include more information regarding the effects of diets on sleep. Especially as you include several different facets (diet quality, timing, and calorie restrictions), which in my opinion are not sufficiently introduced, which makes the choice for this study and setup unclear. Regarding f.e. eat timing, several important papers should be discussed in light of your study: 10.1016/j.cmet.2015.09.005 , 10.1016/j.cmet.2019.11.004

  • I added the informative descriptions in Introduction part with the references follow your recommendation.

Intermittent fasting is a dieting method that controls mealtimes. Low-calorie diets for weight contro l and time-restricted diets using them are commercially available. A time-limited diet, in which a meal was eaten for 10 hours and fasted for the rest of the time, promoted weight loss in patients with metabolic syndrome. Time restricted diet is also known to reduce visceral fat, improve abdominal obesity, lower atherogenic lipids, glycated hemoglobin cholesterol and control high blood pressure. Modified fasting regimens or periodic very low calorie diets have been shown to improve insulin resistance, reduce fasting blood sugar levels, and reduce weight.5

Methods

“Under 20-50 years old” should be changed to “between 20-50 years old”.

--> I modified it according to your recommendation.

What p-value did you consider to be statistically significant?

à In the statistics section, detailed descriptions have been added as follows.

“All values from this study are presented as the mean ± standard variations. The difference between the mean values of the variables including the results of serology, questionnaire and PSG before and after time restricted diet was analyzed using the paired t-test. A p value of less than 0.05 was considered statistically significant. All analyses were conducted using IBM Statistics SPSS software (version 26.0 SPSS, Inc., Chicago, IL, USA).”

In the diet section, please describe what rules the participants needed to follow with regards to the eating time window.

--> In the'Method' section, the diet schedule content has been added as follows.

“Participants were asked to eat for 8 hours from 12 p.m. to 8 p.m. For the rest of the time, he kept fasting, but hunger and thirst were allowed to drink water or Deodeok tea as much as possible. Intermittent coffee was allowed. It was recommended that time restricted diet should be observed on weekends as well.”

Right now, the only focus lays on the low calorie-diet. Also, how did you confirm that the participants adhered to your instructions regarding food amount and food timing?

  • Basically, once a day or more during the week, researchers phoned to check whether they were following the diet well. On the weekends, we didn't check the phone over the phone, and we measured ketone levels every Monday. If you didn't perform well on the weekends, especially if you took food other than the ones provided, your ketone levels were lowered. For this reason, we viewed ketone levels as a basis for indirectly determining whether or not to conduct research.

Participants were found to perform well on weekdays, but the performance rate declined on weekends. Through interviews every week, we classified two or more additional meals on the weekend or two or more meals other than the provided lunch box as an unfaithful participant.

Although not foreseen, we named it a low ketone group because of the low ketone levels in the unscrupulous participant. In other words, the low ketone group is the insincere participant group, and the high ketone group is the faithful participant group.

“During the study period, if the number of meals other than the lunch box provided more than two meals per week was consumed, or the number of additional food intakes more than twice a week during the fasting time, it was defined as insincere participation. Six of the participants did not participate faithfully, and their ketone levels were also low.”

Regarding the polysomnography, where did the participants sleep? And who scored the polysomnography?

--> Before and after the study, they performed polysomnography at the sleep center in the research institute.

 “PSG was conducted before and after forest therapy, but baseline PSG was conducted one week before participation in the sleep lab of the research institute. PSG was performed using a digital PSG machine (Nox A1, Nox Medical Inc., Reykjavik, Iceland). The following variables were monitored: electroencephalogram (EEG; C3-A2, C4-A1, O2-A1, O1-A2), right and left electro-oculogram, submental, both anterior tibialis electromyograms, electrocardiogram, airflow (pressure cannula and thermistor), respiratory effort (piezo-electric bands), oxyhemoglobin saturation (SaO2), and snoring. It was performed by an experienced sleep technician, and the scoring and staging were performed by the sleep physician.”

Results

You mention multiple times that the participants´ body weight and BMI decrease. However, you show no significant differences in these variables when analyzing these measures before and after the intervention. This is certainly a stretch of interpretation, as your results and p-values show  no differences (p=0.539 and p=0.233) Thus, there is no significant weight loss in these participants after the intervention. Please adjust.

  • It is misleading and has been corrected as follows.

“Body weight after this program changed from 82.0±15.6 kg to 78.2±14.1 kg (p=0.539), and BMI decreased from 29.3±4.6 to 27.9±3.8 without statistic significance (p=0.233).”

You say you base the cut off for the ketone values on the highest value of ketone levels. Could you elaborate a bit more on this? Did you take half of the highest value as a cut off? Was this the highest mean value of one participant or of all participants? This is unclear.

  • As mentioned earlier, the program was divided into two groups based on how faithfully they participated in this program. Ketone levels were generally based on known normal, high ketone levels.

In table 3 and 4, is it correct that you only show the statistical outcomes for the high ketone group? Why did you do this?

  • Patients who faithfully followed this program, i.e., the time-restricted diet, had the highest ketone levels at week 2, and showed the high level ranges. Therefore, in the before and after comparison, participants who faithfully performed intermittent fasting showed a large difference in weight, BMI, and body fat compared to those who did not. For that reason, the analysis was conducted in two groups.

Although there are indeed significant changes in insulin in the high ketone group following the intervention (p=0.006), for the insulin resistance (as measured with HOMA), there is only a trend visible (p=0.052), which should be clarified.

  • Thanks for the very important point. I have modified it as follows according to your comment.

“Table 4 shows the results of blood tests in each group. In particular, during the blood test of the high-ketone group, significant changes in insulin level (p=0.006) and an improved trend of insulin resistance as a measured HOMA-IR (p=0.052) were confirmed”

For all tables, please provide an abbreviation list in the description under each table.

à In each table, abbreviations have been resolved and explanatory comments have been added as follows.

AST; Apartate transaminase, ALT; Anine transaminase, BUN; Blood urine nitrogen, r-GTP; Gamma Glutamyl Transpeptidase, ALP; Alkaline phosphatase, TG; Triglyceride, HOMA IR; Homeostatic Model Assessment for Insulin Resistance

REM; Rapid eye movement, AHI; Apnea Hypopnea Index, PLMSi, Periodic limbs movement during sleep index

Also, please specify if these values are mean ± SD, median ± SEM, or ..?

  • We described those values as mean ± SD.

“Fifteen obese volunteers (9 males and 6 females) participated in this study and completed it without failing. The age was 36.8±8.44 (mean ± standard deviation) years old, and the baseline BMI was 29.3±4.62 years old.”

Why did you choose to show each participant´s ketone changes in figure 1 and 2? Given the mean values are already available in the tables, the added value of these figures is unclear to me.

  • It is expressed as a graph to show the overall change. However, as you pointed out, figure 2 is described in the table, and I agree with that there is no need for additional figures in the table as it does not show any particular patterns of change. Figure 2 has been removed.

Please report the outcomes of the Epworth sleepiness scale, Stanford sleepiness scale, and the Korean version of the Pittsburgh Sleep Questionnaire Index, even if they are not significantly different. Also, why did you choose here not to show the overall group findings?

  • It didn't show any significant changes, but I added a table based on your opinion.

Table 5. Questionnaire results before and after high/low ketone group (n=15)

Low Ketone group (n=6)

High Ketone group (n=9)

p

Variables

Before

After

Before

After

PSQI-K 1

STOP-Bang

SSS 2

ESS 3

HADS 4 Anxiety

HADS Depression

ISI 5

1 Pittsburg Sleep Quality Scale-K, 2 Stanford Sleepiness Scale, 3 Epworth Sleepiness Scale, 4 Hospital Anxiety and Depression Scale, 5 Insomnia Severity Index.

As you mention later in your discussion, splitting participants on their ketone body levels is maybe not the right measure to evaluate differences of such a multi-layer intervention.

  • As mentioned earlier, the intervention of several factors cannot be excluded. In particular, the amount of exercise was different for each participant, but the amount of exercise is also a major factor in the change in ketone levels. However, the change in ketone levels was an important and meaningful objective to determine whether thorough dietary control was performed. Because of this, we could not find a more objective indicator than this. Participants were asked to control their diet thoroughly, but there were ethical limits to sanctions.

Although you didn´t evaluate participant´s chronotype, what were their bed and wake times? This could be highly affected by the eating onset and offset, and vice versa. You could use this f.e. from the PSQI. This should also be discussed in the discussion.

  • It was modified as follows to reflect your opinion.

“Third, the chronotypes of the subjects were not reflected in this study. Chrono-types during time restricted diet could be divided into morning and evening types. This program would have been more adaptable in the evening types because of the restricted mealtime. Chronotype could influence the time restricted diets because it is an important factor to determine the beginning and end of the diet. Depending on the chronotype, the success or failure of these dietary restriction studies can vary. Follow-up studies addressing these limitations might be able to resolve any new questions we have gained through this study. “

Discussion

You are testing three different facets of diet (eating time restriction, dietary content (forest products), and calorie restriction) at the same time. How do you know where the (lack of) effects come from? This should be thoroughly discussed. This will also make the comparison to other studies difficult, but this should be at least attempted.

à I fully agree with your opinion. One of the goals of our research was to create a lunch box using forest products, a healthy lunch box based on low calorie, low carb, and high protein. In addition to the effects of a healthy lunch box, the second goal was to determine the effect of weight loss on sleep. So far, many papers have been published that have studied weight loss and described the results. However, sleep studies that have caused this are insufficient. As you commented on, it's hard to see if it's a short-term weight loss effect or whether this is the effect of a timed diet.

However, there are a few things that are clear: the first is that the effects of short-term weight loss have a positive effect on sleep apnea, and the second is that changes in chronobiology caused by a timed diet do not affect sleep structure and sleep time.

“In this study, it's hard to see if it's a short-term weight loss effect or whether this is the effect of a time restricted diet. However, there are a few things that are clear: the first is that the effects of short-term weight loss have a positive effect on sleep apnea, and the second is that changes in chronobiology caused by a time restricted diet do not affect sleep structure and sleep duration. However, it is necessary to confirm how each factor affects through future research.”

The conclusions written here are too bold, especially given non-significance.

à We sympathize with that, and we have revised it more clearly as follows.

“The time-restricted diet using forest products was successful in weight control for a period of 4 weeks and did not change the structure of sleep. The result of weight loss showed a positive effect on improving sleep apnea for 4 weeks. Through this study, we clinically investigated the positive effects of the time restricted diet, and further studies are needed to establish a basis for its mechanism.”

If you choose to keep the ketone-split in your paper, the discussion really needs to include a section where the differences found in high-ketone group but not low-ketone group are discussed.

à According to your opinion, I have described the contents by adding a paragraph in the discussion section as follows. It is noteworthy that Ketone Lebe and sleep have a major link.

Sleep stage N3 (%)

19.73±8.35

21.42±5.80

15.51±3.91

19.09±8.67

“Ketones are one of the indicators in the sleep regulation, and during sleep, ketone bodies in the cerebral fluid increase. Sleep deprivation make increased serum concentration of ketone bodies and increased expression of ketogenesis related genes in brain. The level of ketone bodies and their metabolism in brain could affect sleep homeostasis. It is not clear whether increased ketones play a protective role in sleep or a promotional role through several studies, but it could be seen that it has a positive effect on sleep. In this study, In this study, the rate of increase in slow wave sleep (stage N3) in the high ketone group was greater than that of the low ketone group. Although statistical significance was not shown, further study is needed to determine whether the greater increase in slow wave sleep, that is, deep sleep in the high ketone group, was the result of the sleep-enhancing effect of ketones shown in previous studies on the relationship between ketones and sleep.”

Every person has a different calorie need, also based on f.e. exercise. How could this have impacted the results, as all participants ate 1350 kcals? Or where the participants matched on energy expenditure levels?

  • At the beginning of the study, we provided individual training to all participants. As much as possible, we tried to maintain a similar amount of exercise and lifestyle. Participants were 2 students, 13 were working, and no participant was manual labor. All of them were white collar workers. In addition, there were no shift workers among the participants because shift work affects the sleep awakening cycle.
  • About the decrease in calorie intake per day (as mentioned in the paper), he complained of hunger in the first week (as mentioned in the paper), but it was not intolerable. From the second week, the participants were adjusting to 1350 calories.

yours sincerely, Hyeyun Kim

Round 2

Reviewer 2 Report

Dear authors,

I want to thank you for the thorough answers and clarifications.
I am pleased with the changes you made to the manuscript, and definitely can say that the manuscript improved significantly!

I just have a few small remarks left:

  • You mention in your responses that you ensured participant-adherence to the diet via phone calls. That is great and actually makes the study more robust, and could thus definitely be mentioned in the methods.
  • In line 144 you say: "During the study period, if the number of meals other than the lunch box provided more than 144 two meals per week was consumed, or the number of additional food intakes more than twice a 145 week during the fasting time, it was defined as insincere participation. Six of the participants did 146 not participate faithfully, and their ketone levels were also low." This is a bit confusing. If I understood what you wanted to say correctly, I would change it to:
    "During the study period, if the participants consumed more than two meals per week that where not provided in the lunchbox, or if the participant ate food during the fasting time (between XX and XX) for more than twice per week, this was defined as unadequate participation. Six of the participants did not adhere to our instructions, which was also represented with their low ketone levels"
  • In line 150 you write that "Figure 1 shows the 150 changes in ketone levels performed at 1-week intervals during the study period in the high and low 151 ketone groups. " I suppose this should be Figure 2.
  • Not related to the paper, but your reviewer can be either male or female :)

Best of luck!

Author Response

Dear. reviewer

I appreciated to your kind and detailed comments for developing my article.

  • You mention in your responses that you ensured participant-adherence to the diet via phone calls. That is great and actually makes the study more robust, and could thus definitely be mentioned in the methods.

à Thanks for your comment. We have added the following for our phone contact.

“ We had called the participants at every morning to make sure they have eaten only the supplied meal box and kept the designed time according to the program, and encouraged them to faithfully carry out the program on that day.”

  •  
  • In line 144 you say: "During the study period, if the number of meals other than the lunch box provided more than two meals per week was consumed, or the number of additional food intakes more than twice a week during the fasting time, it was defined as insincere participation. Six of the participants did not participate faithfully, and their ketone levels were also low." This is a bit confusing. If I understood what you wanted to say correctly, I would change it to: 
    "During the study period, if the participants consumed more than two meals per week that where not provided in the lunchbox, or if the participant ate food during the fasting time (between XX and XX) for more than twice per week, this was defined as unadequate participation. Six of the participants did not adhere to our instructions, which was also represented with their low ketone levels"

àI will modify it according to your instructions. I think that what we want to talk about in our study is better communicated.

" During the study period, if the participants consumed more than two meals per week that where not provided in the lunchbox, or if the participant ate food during the fasting time (between 12 p.m. and 8 p.m.) for more than twice per week, this was defined as unadequate participation. Six of the participants did not adhere to our instructions, which was also represented with their low ketone levels. "

  • In line 150 you write that "Figure 1 shows the 150 changes in ketone levels performed at 1-week intervals during the study period in the high and low 151 ketone groups. " I suppose this should be Figure 2.

à I modified it according to your instructions.

“Figure 2 shows the changes in ketone levels performed at 1-week intervals during the study period in the high and low ketone groups.”

  • Not related to the paper, but your reviewer can be either male or female :)

à I praise you for your wit. I can tell you your gender.

Thank you again for your sincere, witty, detailed and kind comments.

With COVID19, these studies are now constrained. It is unfortunate that it has become difficult to do further follow-up research. Nevertheless, I have met a great reviewer like you and have enjoyed editing the paper over the past few weeks. Thanks to you, I was able to be born again with a great eye.

Best of luck!
